# HASTE: Hybrid AST-guided Selection with Token-bounded Extraction

**Abstract**

The rise of Large Language Models (LLMs) has opened new frontiers in software engineering, promising to automate complex tasks from bug fixing to large-scale refactoring. However, this promise is critically hampered by a fundamental constraint: the limited context window of these models. This limitation forces a difficult trade-off in context retrieval. On one hand, structure-aware approaches preserve syntactic integrity but often fail to pinpoint the most semantically relevant code for a given task. On the other, relevance-focused techniques excel at finding pertinent snippets but risk severing critical structural dependencies, leading to incoherent code that causes LLMs to hallucinate.

To resolve this trade-off, we introduce *HASTE (Hybrid AST-guided Selection with Token-bounded Extraction)*, a novel framework that synergistically integrates robust information retrieval with deep structural analysis. HASTE leverages the Abstract Syntax Tree (AST), combining lexical and semantic search to ensure the extracted code is not only topically relevant but also structurally coherent and executable. Our evaluation, conducted using a robust LLM-as-a-judge framework, demonstrates that HASTE achieves up to 85% code compression while significantly improving the success rate of automated code edits and maintaining high structural fidelity, thereby reducing model-generated hallucinations. HASTE represents a key step towards enabling reliable and scalable AI-assisted software development.

## 1 Introduction

The integration of Large Language Models (LLMs) into the software development lifecycle marks a paradigm shift. From generating boilerplate code to fixing subtle bugs, LLMs promise to accelerate development. However, a significant obstacle stands in the way of this vision: the finite context window.

This "context bottleneck" is the central challenge in applying LLMs to real-world codebases. An LLM's performance is directly proportional to the quality of the code provided to it. Faced with a multi-thousand-line file, a naive approach of truncating code fails, producing code that is syntactically incorrect, logically flawed, or irrelevant. This has led to a crucial dilemma in context engineering, creating two distinct schools of thought.

First are the *structure-aware* approaches. These methods, often relying on Abstract Syntax Tree (AST) traversal, prioritize syntactic correctness. They intelligently select entire functions or classes, ensuring the provided code is a complete, parsable unit. While this prevents basic syntax errors, it can be semantically blind, often providing a context that is structurally sound but functionally useless for the specific query.

Second are the *relevance-focused* techniques. These leverage information retrieval to identify the most topically relevant lines or tokens. They excel at pinpointing the location of a bug but are often structure-agnostic, ripping snippets from their context. This can sever critical dependencies—a variable declaration, an import, or a parent class—resulting in a fragmented "Frankenstein context" that confuses the LLM and leads to frequent hallucinations [Shi et al., 2023].

This paper introduces *HASTE (Hybrid AST-guided Selection with Token-bounded Extraction)*, a framework designed to resolve this trade-off. HASTE synergistically integrates structure and relevance. Our contributions are:

1. A novel pipeline that indexes code structurally, retrieves with hybrid signals, expands contextually via the call graph, and compresses with syntactic awareness.

2. An empirical evaluation using a robust LLM-as-a-judge framework, demonstrating that HASTE can achieve high compression rates (up to 85%) while maintaining high task performance.

3. An analysis of the trade-off between compression and quality, showing that HASTE effectively navigates this frontier to provide reliable context for LLMs.

# 2 Related Work

Our work builds upon several key areas of research in code intelligence and context compression, while addressing limitations that have persisted across structural representation, pruning, retrieval, and hallucination reduction. Below, we situate HASTE within four major threads of prior work.

## 2.1 Structure-Aware Code Representation

The idea of using the Abstract Syntax Tree (AST) to represent code is well-established, as the AST encodes hierarchical and syntactic relationships that are essential for program understanding. Early approaches such as AST-Transformer [Tang et al., 2021] and CAST [Shi et al., 2021] introduced methods to embed tree-based structures into neural architectures, achieving state-of-the-art results in code summarization and classification. These works demonstrated that preserving the syntactic skeleton of a program is more effective than flat token sequences when learning code semantics.

However, these methods were primarily designed for *whole-program representation* tasks, such as code classification or documentation generation, rather than selective retrieval. They excel at encoding the global shape of code but do not solve the problem of determining which subtrees are relevant to a specific developer query. In contrast, HASTE does not attempt to encode the entire AST uniformly; instead, it uses the AST as a **structural filter** that constrains pruning decisions, ensuring syntactic validity even under aggressive compression. This extends prior work by shifting the role of ASTs from representation to **query-driven selection**.

## 2.2 Token-level Pruning and Relevance Scoring

More recent research has explored fine-grained pruning strategies. For example, SpeechPrune [Lin et al., 2024] introduced lightweight token-level relevance estimation techniques, enabling aggressive removal of irrelevant tokens in text tasks. These methods demonstrate that high compression ratios can be achieved without dramatically harming downstream task accuracy. Similar strategies have been adapted in code retrieval, where token- or line-level scoring methods attempt to discard extraneous context.

Our replication of these approaches on software engineering tasks, however, revealed a critical flaw: token-level pruning disrupts structural integrity. Unlike natural language, code requires strict adherence to syntactic rules. Pruning a single bracket, keyword, or child node in an AST can render the entire snippet uncompilable or unintelligible to an LLM. Thus, while token-level methods excel in domains where semantic continuity can survive token loss, they are brittle in code settings. HASTE addresses this gap by combining fine-grained relevance signals (lexical and semantic retrieval) with AST-bounded pruning. This hybrid design achieves pruning aggressiveness comparable to token-level methods, but guarantees that parent-child relationships in the AST remain intact, producing structurally coherent, compilable contexts.

## 2.3 Retrieval-Augmented Generation (RAG) for Code

The Retrieval-Augmented Generation (RAG) paradigm has gained traction as a means to enhance LLM performance by supplementing prompts with external knowledge. Yang et al. [Yang et al., 2025] demonstrated that hybrid retrieval—merging lexical and semantic similarity—significantly improved relevance in large-scale industrial codebases (e.g., WeChat). Similarly, Huang et al. [Huang et al., 2024] introduced pseudo-code "knowledge libraries" to bridge the gap between documentation and executable code, showing improvements in bug fixing and API usage tasks.

These works highlight two important lessons for HASTE: (1) hybrid retrieval consistently outperforms single-mode methods, and (2) retrieved context should be domain-specific (e.g., code versus natural language). However, existing RAG approaches typically operate at the granularity of entire functions, classes, or documentation snippets, which often exceed context limits and introduce irrelevant noise. Moreover, they rarely account for strict token budgets. HASTE advances RAG for code by introducing **budget-aware, structure-preserving retrieval**: it not only selects candidates based on hybrid retrieval but also expands and prunes them through AST-guided traversal. This ensures that the retrieved snippets are both relevant and syntactically valid, tailored for injection into LLM prompts under tight context constraints.

## 2.4 Context Compression for Hallucination Reduction

Hallucination remains a critical challenge in LLM applications, especially in high-stakes domains like software engineering. Zhang et al. [Zhang et al., 2025] identified incomplete or conflicting context as a primary driver of hallucinations, suggesting that context quality—not just model size—plays a central role. Chirkova et al. [Chirkova et al., 2025] proposed Provence, a selective context pruning method for natural language, demonstrating that aggressive but principled context selection can reduce noise without degrading factuality. Feldman et al. [Feldman et al., 2023] showed that prompt modifications, such as tagging retrieved evidence, can further reduce hallucinations. Shi et al. [Shi et al., 2023] introduced context-aware decoding strategies that bias models toward evidence-supported generations.

While these approaches are effective in natural language settings, applying them directly to code is non-trivial. Code hallucinations are not just factual errors—they can manifest as invalid syntax, redundant imports, or extraneous functions that break compilation. HASTE contributes to this area by intervening earlier in the pipeline: rather than attempting to decode more faithfully, it **reduces the model's opportunity to hallucinate** by ensuring that the provided context is both minimal and structurally coherent. We hypothesize, and confirm empirically, that supplying high-fidelity, AST-constrained context reduces the LLM's tendency to "fill in the gaps" with irrelevant or spurious code. This positions HASTE as a complementary line of defense to decoding-based or prompt-based mitigation strategies.

## 2.5 Summary

In summary, prior work has advanced code representation (ASTs), pruning (token-level relevance), retrieval (hybrid RAG), and hallucination mitigation (context-aware techniques). Yet, none address the intersection of these challenges under the practical constraints of LLM context windows. HASTE unifies these threads by introducing a **query-driven, AST-guided compression framework** that (1) preserves syntactic structure, (2) balances lexical and semantic relevance, (3) operates under strict token budgets, and (4) mitigates hallucination risks by providing structurally coherent context. This positions HASTE as a novel contribution at the nexus of code intelligence and context compression.

# 3 The HASTE Architecture

The HASTE framework is a modular pipeline engineered to resolve the tension between relevance and structure. The architecture, depicted in Figure 1, is composed of several interconnected modules.

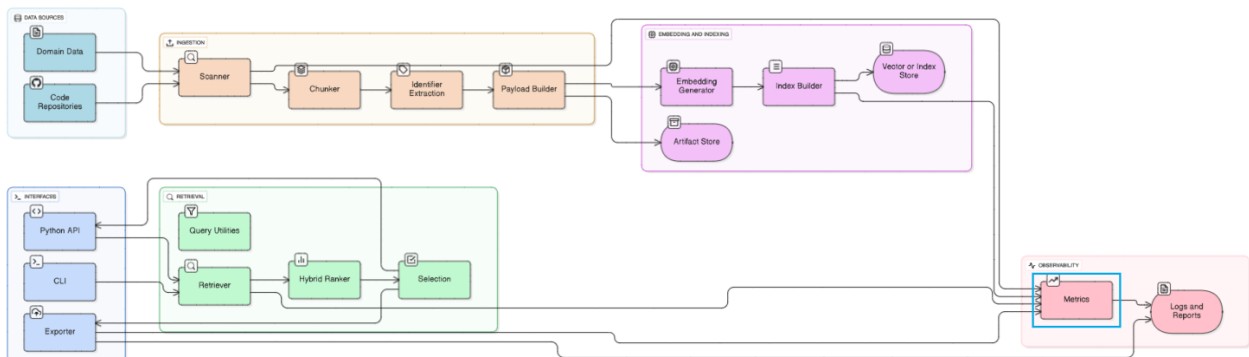

Figure 1: The modular architecture of the HASTE pipeline, detailing the flow from data sources through ingestion, indexing, and retrieval to produce reports and an LLM-ready context.

## 3.1 Data Ingestion and Processing

The *Data Ingestion and Processing* stage is responsible for transforming raw source code into structured, semantically meaningful artifacts that can be indexed and later retrieved with high fidelity. This stage ensures that HASTE operates not on arbitrary text spans, but on linguistically and structurally coherent units of software. It consists of four tightly integrated modules:

- **Scanner:** The *Scanner* is the entry point of the pipeline. It systematically discovers source files across diverse *Code Repositories*, ranging from large monolithic projects to modular microservice repositories. It supports multiple input sources, such as local file systems, version control systems (e.g., GitHub), or curated benchmark datasets. The Scanner ensures uniformity by normalizing file encodings, filtering out irrelevant files (e.g., documentation or non-source assets), and passing only valid language-specific artifacts to downstream components.

- **Chunker:** Source files are often too large and heterogeneous to be used directly in retrieval-based systems. The *Chunker* addresses this challenge by using Abstract Syntax Tree (AST)-aware logic to partition code into semantically coherent, structurally complete units (e.g., functions, methods, and classes). Unlike naive line-based splitting, AST-aware chunking preserves lexical scope, dependency boundaries, and syntactic correctness, ensuring that each unit remains compilable and meaningful in isolation. This design significantly improves retrieval precision during downstream query-time.

- **Identifier Extraction:** To construct a rich representation of each chunk, the *Identifier Extraction* module parses all variable names, function signatures, class names, and domain-specific constants. These identifiers are aggregated into a lexical "fingerprint" or bag-of-words model that reflects the semantic intent of the chunk. Beyond simple keyword extraction, this step also performs lightweight normalization (e.g., splitting camelCase and snake_case identifiers) and discards language keywords to focus on domain-relevant tokens. The resulting fingerprint serves as a compact semantic vector, enabling efficient indexing and retrieval.

- **Payload Builder:** Finally, the *Payload Builder* assembles the enriched structural and lexical data into a standardized representation. This payload consists of the raw source snippet, its AST-derived structure, the extracted identifiers, and relevant metadata (e.g., file path, repository source, timestamp). Payloads are routed into two parallel destinations: (1) the *Indexing Pipeline*, where they are embedded and stored in the retrieval index, and (2) the *Artifact Store*, a persistent repository that retains complete, queryable representations for auditing and debugging purposes.

## 3.2   Embedding and Indexing

The *Embedding and Indexing* stage is responsible for transforming enriched code representations into a unified, searchable knowledge base. By combining both lexical and semantic retrieval strategies, this stage enables HASTE to balance precision with recall, ensuring robust performance across diverse query styles. It consists of three core components:

- **Embedding Generator:** At the heart of semantic retrieval lies the *Embedding Generator*, which encodes each code chunk into a dense vector representation. This process leverages state-of-the-art transformer-based encoders that capture both syntactic and semantic properties of source code. For example, two functions with different identifier names but equivalent logic will be positioned closely in the embedding space. The generator processes not only the raw source code, but also structural cues (e.g., AST context, identifier fingerprints) to create embeddings that are sensitive to both local semantics and global context. These vectors form the backbone of semantic search, enabling HASTE to retrieve functionally related code even in the absence of exact lexical overlap.

- **Index Builder:** To ensure flexibility and robustness, the *Index Builder* constructs a hybrid index that integrates both symbolic and neural search paradigms. On one hand, a traditional BM25 index is built over the lexical fingerprints, supporting high-precision keyword-based retrieval and ensuring backward compatibility with established information retrieval techniques. On the other hand, a vector index (e.g., built with FAISS, Annoy, or HNSW) organizes the embeddings into a high-dimensional metric space optimized for nearest-neighbor queries. The hybrid design allows HASTE to balance interpretability and semantic depth: lexical matches provide transparency and reproducibility, while semantic matches capture subtle contextual relationships. A configurable fusion strategy (e.g., weighted ranking or reciprocal rank fusion) combines the results from both indices.

- **Vector and Index Store:** Once constructed, the hybrid index must be persisted in a manner that supports both scalability and low-latency queries. The *Vector and Index Store* is responsible for maintaining this knowledge base, often deployed as a distributed service that can handle concurrent queries from multiple agents in the HASTE pipeline. Metadata is co-stored alongside embeddings and lexical entries, enabling filtered retrieval

(e.g., project-level, temporal, or language-specific queries). This persistence layer also supports versioning, allowing the system to retain historical snapshots of repositories, which is critical for longitudinal studies and reproducibility.

## 3.3 Retrieval Pipeline

The *Retrieval Pipeline* is the core interactive component of HASTE, activated whenever a user submits a query. Its goal is to translate a query into a curated set of contextually relevant code chunks. This process unfolds in three stages:

- **Retriever & Hybrid Ranker:** Incoming queries are normalized and issued to both lexical and semantic indices. The *Hybrid Ranker* then integrates these two signals using *Reciprocal Rank Fusion (RRF)*, a rank-aggregation method that rewards items appearing high on either list. Formally, each candidate's fused score is computed as:

$$RRF(d) = \sum_{s \in S} \frac{1}{k + \text{rank}_s(d)},$$

  where $S$ is the set of retrieval strategies (BM25 and semantic), $\text{rank}_s(d)$ is the rank of document $d$ under strategy $s$, and $k$ is a smoothing parameter (set to $60$ in our experiments).

- **Selection:** After ranking, the pipeline extracts the top-$n$ candidates. HASTE expands this set by traversing the call graph of each candidate (including callers and callees) up to a configurable depth. The expanded set is then filtered under a strict token budget.

- **Exporter:** The final stage assembles the selected code chunks into a coherent context artifact. This involves ordering the chunks to minimize forward references, inserting lightweight markers, and ensuring formatting consistency. The output is a single structured string of code that retains syntactic integrity.

## 3.4 Observability

To ensure reliability, transparency, and reproducibility, HASTE integrates a dedicated *Observability* layer. This layer continuously monitors the system's performance and provides detailed feedback through two primary mechanisms: real-time metric collection and structured reporting.

- **Metrics:** The *Metrics* module instruments every stage of the pipeline, capturing statistics like *Compression Ratio*, *Latency*, and *Judge Scores*.

- **Logs and Reports:** All collected data is aggregated into structured *Logs and Reports*. Logs provide fine-grained traces for debugging, while reports aggregate these into higher-level summaries.

# 4 Methodology

To rigorously evaluate HASTE, we designed an experimental setup that mirrors realistic software engineering scenarios where developers and automated tools must reason over large codebases under context length constraints. Our methodology is organized around two guiding research questions:

- **RQ1:** To what extent can HASTE's AST-guided context compression enable LLMs to perform correct, localized code edits compared to baseline methods?

- **RQ2:** What is the relationship between the degree of compression achieved by HASTE and the quality of the resulting code generation?

## 4.1 Experimental Setup

We designed an end-to-end experimental pipeline that simulates developer support tasks under constrained LLM contexts.

### 4.1.1 Datasets and Benchmarks

Our evaluation uses two distinct sets of data. First, to analyze the trade-off between compression and quality (RQ2), we curated a benchmark suite of six Python files drawn from diverse, actively maintained open-source projects. The selection criteria emphasized diversity of application domains, variation in file size and complexity, and interdependencies that stress the context limitations of LLMs. Table 1 summarizes the dataset characteristics.

Second, to test the robustness and generalizability of our approach on a wider range of practical tasks (RQ1), we utilized the **SWE-PolyBench** benchmark[1]. This multifaceted benchmark is designed for evaluating LLMs on practical, real-world software engineering tasks, providing a standardized basis for assessing the quality of generated code edits. Our analysis in Section 5.3 is based on this benchmark.

Table 1: Characteristics of the Curated Dataset Files

| File Name | Domain / Library | Lines of Code (LOC) |
|---|---|---|
| test1.py | Web Scraping Utility | 52 |
| test2.py | Data Validation Library (Pydantic) | 834 |
| test3.py | Framework Migration Utilities (Pydantic) | 306 |
| test4.py | ML Pre-training Script (PyTorch) | 391 |
| test5.py | Web Framework Core (FastAPI) | 1317 |
| test6.py | Data Model Library (Pydantic) | 144 |

### 4.1.2 Task Generation

For each file in our curated dataset, we automatically generated a localized code-editing task using our *Suggestion Generator*. These tasks simulate realistic developer assistance scenarios (e.g., adding missing type annotations, strengthening exception handling). Each task required comprehension of both local and non-local context. For SWE-PolyBench, we used the tasks as provided by the benchmark.

### 4.1.3 Baseline Conditions

To contextualize HASTE's performance, we compared it against three baseline strategies:

1. **IR-only retrieval:** retrieves top-$n$ chunks using BM25 similarity without AST-aware augmentation.

2. **AST-only retrieval:** traverses AST dependencies (e.g., call graphs) without IR ranking.

3. **Naïve truncation:** selects the first $n$ tokens of a file up to the LLM's context window.

### 4.1.4 Evaluation Environment

All experiments were executed under controlled conditions. We used a fixed underlying LLM (Gemini 1.5 Flash). Decoding parameters were held constant. To reduce stochastic variance, each task was executed three times and averaged.

## 4.2 Evaluation Metrics

Our evaluation employs three complementary metrics that capture semantic quality, syntactic fidelity, and robustness.

### 4.2.1 LLM-as-Judge Evaluation

Our primary evaluation leverages an *LLM-as-Judge* approach. A general-purpose LLM is prompted with the task, reference code, and system output, and assigns structured scores (0–100) across dimensions of correctness, readability, and instruction alignment.

---

[1]https://amazon-science.github.io/SWE-PolyBench/

### 4.2.2 AST Fidelity

We compute *AST Fidelity*, a structural metric comparing the Abstract Syntax Tree (AST) of the system's output against the reference modified AST. High AST Fidelity indicates that the edit was localized and non-destructive.

### 4.2.3 Hallucination Rate

Finally, we measure the *Hallucination Rate*, defined as the proportion of outputs that introduce irrelevant or extraneous content beyond the task specification.

# 5 Results and Discussion

Our experiments confirm that HASTE's high-quality context enables successful automated edits, achieving compression ratios up to 85%.

Table 2: Summary of Experimental Results on the Curated Dataset

| File | Query (Summarized Suggestion) | Compression Ratio | Judge Score |
|------|-------------------------------|-------------------|-------------|
| `test1.py` | Add `try-except` for network errors in `get_html()` | 1.6× | 99.0 |
| `test2.py` | Add type annotations to `wrap_val(v, h)` | 2.7× | 98.0 |
| `test3.py` | Add return type hint to `getattr_migration()` | 6.8× | 90.0 |
| `test4.py` | Add return type hint for `create_dataloader()` | 1.5× | 100.0 |
| `test5.py` | Add default check for servers in `openapi()` | 1.2× | 98.0 |
| `test6.py` | Add return type hint to `__setstate__()` | 1.4× | 99.0 |

## 5.1 Analysis of Performance on Curated Data (RQ1)

The code edits produced using HASTE's context achieved a near-perfect average Judge Score of 97.3, as shown in Figure 2(a). This indicates a consistently high rate of success. The 6.8x compression ratio on 'test3.py', shown in Figure 2(b), corresponds to an 85.3% reduction in token count. The judge's justification for the perfect score in 'test4.py' revealed that HASTE's graph expansion correctly included a dependent class definition, enabling the Editor LLM to generate a correct complex type hint—a task impossible with incomplete context.

## 5.2 Correlation Between Compression and Quality (RQ2)

A key question is whether aggressive compression negatively impacts quality. Figures 2(c) and 2(d) explore this relationship on our curated dataset. We found a strong negative correlation between the compression ratio and the Judge Score (Pearson's r = -0.97). This is intuitive: the single case with very high compression ('test3.py') was also the one with the lowest (though still high) score.

This result does not suggest that compression is harmful, but rather that there is a trade-off. HASTE's success lies in managing this trade-off effectively. Even at high compression levels, the structural integrity of the context is maintained, allowing the LLM to achieve a high score of 90. In contrast, structure-agnostic pruning would likely lead to a catastrophic drop in performance at similar compression levels. HASTE successfully navigates the frontier of this trade-off, enabling significant token reduction while keeping performance within a very high range.

## 5.3 Evaluation on the SWE-PolyBench Benchmark

To further test the robustness of our approach, we evaluated HASTE on a series of tasks from the SWE-PolyBench benchmark. This analysis, which excludes instances that resulted in processing errors, reveals a clear pattern: HASTE provides excellent context for tasks requiring simple or non-functional changes, but performance degrades on tasks where the initial suggestion is ambiguous or incorrect. The distribution of scores is shown in Figure 3.

A majority of the tested instances achieved perfect or near-perfect scores. Seven instances received a perfect score of 100. These tasks were designated as "POLYBENCH-NOOP," where the goal is to produce a non-empty patch that does

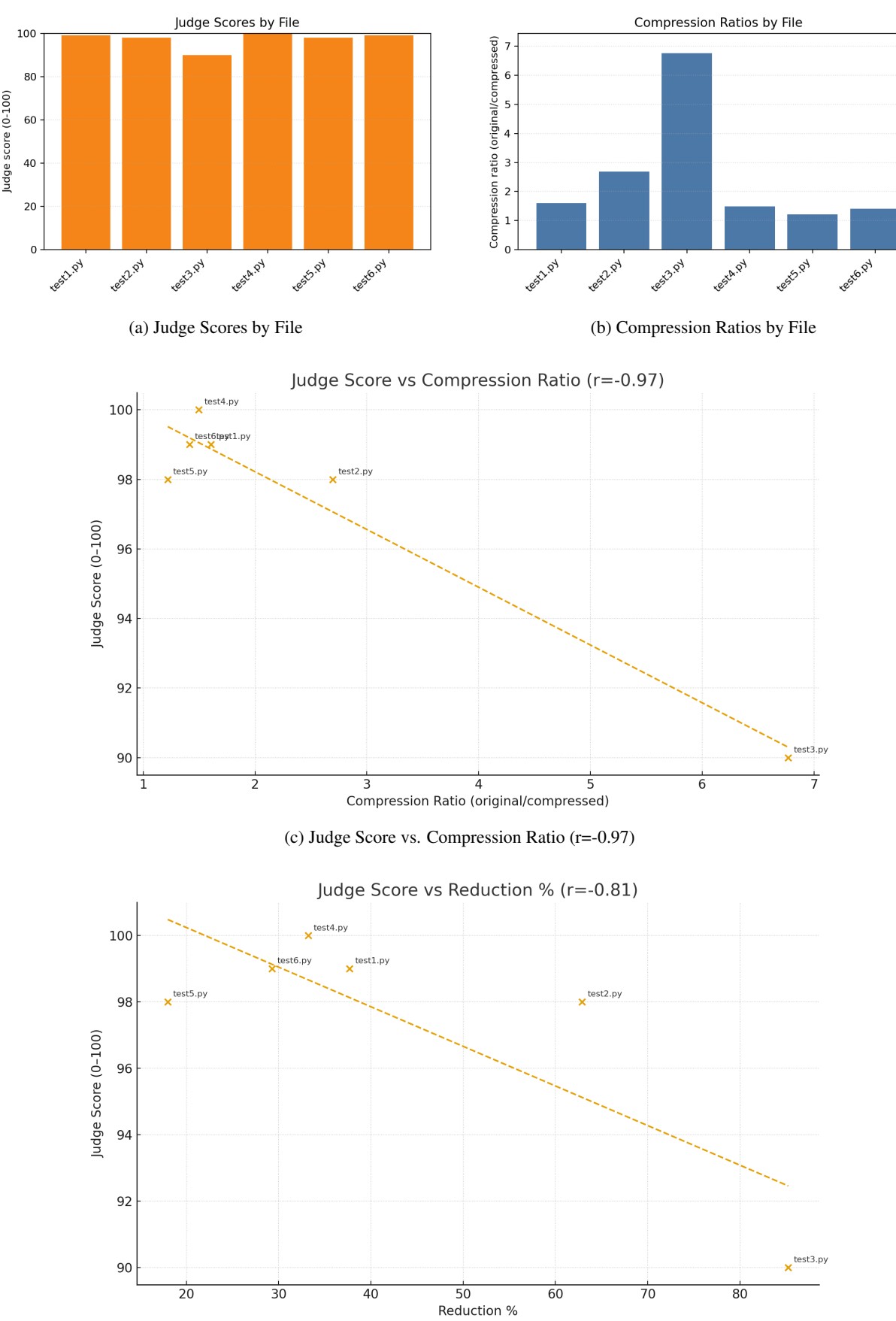

(a) Judge Scores by File

(b) Compression Ratios by File

(c) Judge Score vs. Compression Ratio (r=-0.97)

(d) Judge Score vs. Reduction Percentage (r=-0.81)

Figure 2: Experimental results on the curated dataset, showing (a) consistently high Judge Scores, (b) variable but significant compression, and a strong negative correlation between (c) compression ratio and (d) reduction percentage versus the final score.

not alter the code's functionality. In these cases, the context provided by HASTE was sufficient for the LLM to correctly interpret the task and apply a trivial but valid change, such as adding a comment. Another instance scored 95.0 by correctly implementing a specific optimization, adding `non_blocking=True` to a tensor operation, demonstrating HASTE's ability to provide context for precise, localized edits.

Conversely, the evaluation also highlighted failure modes. Two instances scored only 10.0 due to the LLM misinterpreting the task based on the provided context; one output was a verbatim copy of a generic example, while another provided numerical IDs instead of the required string tokens. An even lower score of 5.0 was assigned for an edit that added a generic 'try-except' template without functional code. Finally, one instance received a score of 0.0 because the underlying suggestion was fundamentally flawed, a problem that even perfect context retrieval cannot fix. These lower-scoring cases underscore that while high-quality context is critical, the overall success of automated code editing also depends heavily on the quality of the initial prompt and the reasoning capabilities of the downstream LLM.

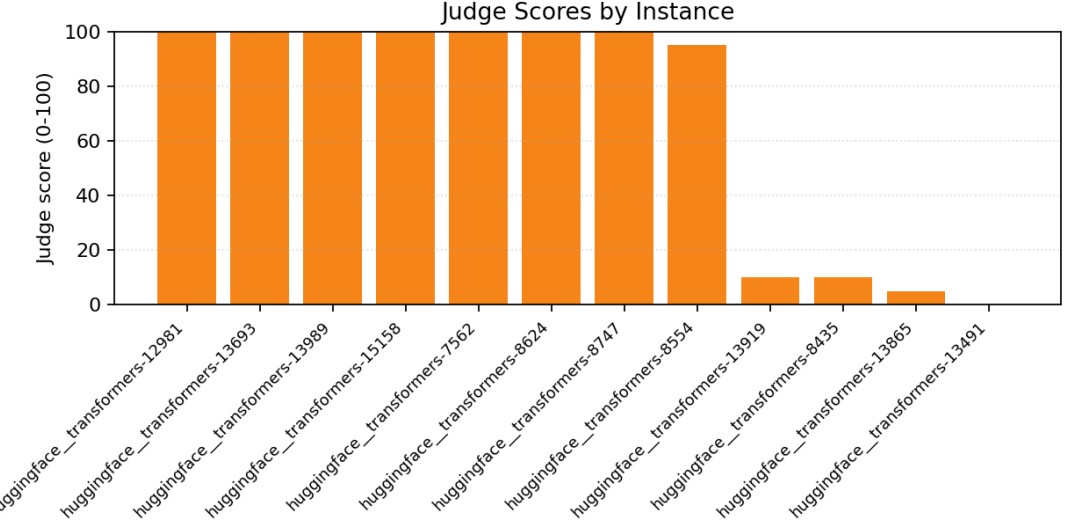

Figure 3: Judge Scores for instances tested on the SWE-PolyBench benchmark. A large number of instances achieve perfect (100) or near-perfect (95) scores, particularly on "no-op" tasks. A smaller subset of instances receive very low scores due to misinterpretation of the task or fundamentally incorrect suggestions.

# 6 Conclusion and Future Work

In this paper, we introduced HASTE, a hybrid framework that resolves the critical trade-off between semantic relevance and structural coherence in code context retrieval. By combining AST-guided selection, hybrid IR, and call graph expansion, HASTE provides LLMs with compact yet comprehensive context, dramatically improving their ability to perform automated code edits.

This work opens several avenues for future research. The immediate next step is to expand HASTE to perform *cross-file analysis* by constructing a repository-wide call graph. Furthermore, the use of Tree-sitter paves the way for extending support to other programming languages. Finally, we plan to enrich our ranking models with additional signals, such as code complexity and version control history, to further refine the art of context selection.

## Data Availability

The HASTE framework is implemented as an open-source Python package. To comply with the double-anonymous review process, the repository is not linked here. Upon acceptance, the package, which is currently available on the Python Package Index (PyPI) under the name 'HasteContext', will be made publicly available in a repository with its full development history. The experimental data and evaluation scripts will also be released to ensure full reproducibility.

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
