# OpenReview forum: "HASTE: Hybrid AST-guided Selection with Token-bounded Extraction"
_ICLR.cc/2026/Conference — ICLR 2026 Conference Withdrawn Submission_

### Official Review · Reviewer_Wkb8 · 2025-11-01

**Soundness:** 1
**Presentation:** 2
**Contribution:** 2
**Rating:** 2
**Confidence:** 4

**Summary:**

The authors propose using AST information to ensure the correct syntax of code snippets retrieved as additional context for LLMs during code tasks. They also ensure that the syntax remains valid during context compression, thereby further enhancing the usefulness of the retrieved information. They validate the approach, which they name HASTE, on a curated dataset and on SWE-Polybench. They demonstrate the performance in terms of LLM-as-judge and compression ratio.

**Strengths:**

- Proposes syntax-aware data retrieval

**Weaknesses:**

- The paper seems to confuse what syntax and structure either in the description of the method, or for the method itself (the writing is not clear enough to allow this reviewer to distinguish these cases)
- It remains unclear why LLMs require valid syntax rather than being resilient to syntax noise
- The LLM-as-judge scores are not grounded in manual analysis (especially given the small dataset for the curated case)
- The LLM-as-judge has scores across multiple dimensions; only the aggregate score is reported
- Paper format issues: Figure 1 is a raster image and of low quality, making reading difficult; the paper does not seem to use the ICLR paper template.

**Questions:**

Q1: How did you select the curated dataset?
Q2: What licenses do you allow in the data ingestion? How do you ensure compliance with software licenses?
Q3: How do you assess the hypothesis that LLMs are vulnerable to syntax noise? (There is a naive truncation baseline, but the results seem unreported)

**Details Of Ethics Concerns:**

It is unclear if the authors perform ethical crawling (is the data ingested permissively licensed, and does the data allow use for AI training).
It is unclear if the authors allow a mechanism for opt-out from crawling.
The curated dataset has missing details: what projects? Under what license? How were they selected?

---

### Official Review · Reviewer_6h7r · 2025-11-01

**Soundness:** 1
**Presentation:** 1
**Contribution:** 1
**Rating:** 0
**Confidence:** 5

**Summary:**

This paper proposes HASTE, a code context retrieval framework that attempts to exploit structural information from Abstract Syntax Trees (AST) and combine it with semantic search from code. This architecture aims to address the context window limitations of LLMs when handling large codebases.

This work includes a minimal evaluation on 6 undisclosed Python files. Here the context compression is quantified by measuring the quality of “automated edits” listed in Table 2 (there are no details on these tasks). The abstract and introduction are stating that the approach yields up to 85% rate of context compression, yet it is not clear how good this is since there is no quantitative comparison to other state-of-the-art techniques.

Overall, a lot of technical details are left out (on methods and evaluation) and there is no appendix or supplementary materials to understand them better.

**Strengths:**

- The problem of context compression for code-related tasks is still relevant despite of the very large context window sizes of recent LLMs.
- The idea of combining semantic relevance and structural coherence has some merit and was already exploited successfully in pre-LLM research, e.g. https://arxiv.org/abs/1910.00577 .

**Weaknesses:**

- The overall quality of the presentation, evaluation, and related work of this submission is way below the standards of ICLR or even most 2nd tier conferences. Lack of essential information makes it impossible to judge the conceptual contribution adequately. There are too many issues to improve so the following is just a sample of most striking problems.
- The paper lacks even basic information on the approach (e.g. the key parts like the “Embedding Generator” and “Index Builder” in Sec. 3.2). Also, most other aspects are not sufficiently explained, for example the evaluation metrics (Sec. 4.2), the “automated edits” tasks, test files and other data, parameter settings (e.g. the LLM employed as a Judge), and other aspects crucial for ensuring the reproducibility of the results.
- Although the paper positions the trade-off between semantic relevance and structural coherence as its central focus, it fails to clarify how HASTE quantifies this trade-off at the methodological level, leaving its main contribution at a conceptual level.
- The chosen baseline methods (e.g., IR-only, AST-only, naive truncation) are overly simplistic and do not compare against state-of-the-art approaches, failing to convincingly demonstrate the true advantages of the proposed framework. There is no comparison to prior state-of-the-art work.
- For failure cases in the experiments, the paper simply attributes them to “fundamentally flawed initial suggestions” or “LLM misinterpreting the task,” without critically examining the quality of its retrieved context, resulting in insufficient analytical depth.
- The benchmark used to address the core research question (RQ2) consists of only six Python files. Conclusions drawn from such a small sample are unreliable.
- The paper does not follow the formatting guide of ICLR and has partially bad structure (e.g. in Introduction).

**Questions:**

- How does this approach compare to state-of-the-art baselines?
- Clarify how HASTE quantifies and manages the trade-off between semantic relevance and structural coherence.
- Provide the missing details, including the methodology, the Judge-LLM model, evaluation prompts, and key parameter settings, etc.

---

### Note · Authors · 2025-11-12

I have read and agree with the venue's withdrawal policy on behalf of myself and my co-authors.